# An Asynchronous Multi-Agent Framework for Adaptive Tool-calling Data Synthesis

## Abstract

The efficacy of Large Language Model (LLM) agents is fundamentally determined by the quality of tool-calling training data. However, current synthesis paradigms mainly rely on successful-only trajectories and static pipelined workflows. This approach not only makes agents prone to runtime crashes, but also fails to cover many unexplored scenarios. To bridge this gap, we propose **AMAS**, a hierarchical and Asynchronous Multi-Agent data Synthesis framework based on a bi-level architecture. Our framework decouples centralized planning from distributed execution, supporting dynamic task scheduling and accommodating trials and errors inherent in realistic tool-calling tasks. As a result, this adaptive mechanism transforms tool execution exceptions which are traditionally discarded as waste into robust self-correction training signals. Comprehensive evaluations show that our method simultaneously enhances data robustness and diversity, context efficiency, and synthesis quality compared to source baselines. Empirically, fine-tuning Qwen3-4B with synthetic data yields consistent improvements across BFCL and $\tau^2$-bench. We hope this work will help inspire community to explore the boundaries of LLM agentic data progress.

## 1. Introduction

The rapid evolution of Large Language Models has marked a transition from passive text generators to proactive agents capable of utilizing external tools to solve complex tasks (Talebirad & Nadiri, 2023; Wu et al., 2024; Guo et al., 2024). The proficiency of such tool-calling capabilities heavily relies on the availability of refined data substrates (Albalak

et al., 2024). However, a substantial existing datasets reveal a pervasive successful-only trajectory bias (Shi et al., 2025a; Jiang et al., 2025). Most open sourced datasets prioritize flawless execution paths and complex error handling processes in the real world are inefficiently covered (Kokane et al., 2024; Kumar et al., 2024; Audran-Reiss et al., 2025). Consequently, models trained on such idealized datasets might struggle to generalize, lacking the self-correction capabilities required to resolve inevitable API exceptions or logical anomalies during deployment and cause hallucinations (Bandi et al., 2025; Sriramanan et al., 2024; Zhang et al., 2025b; Xu et al., 2024).

Despite recent efforts in data synthesis, we identify two fundamental limitations in current methodologies that potentially hinder the development of robust agents: **(1) static pipeline and information loss**. Prevalent synthesis methods typically take a paradigm of generation followed by filtering (Pandit et al., 2025; Wang et al., 2025b), which is computationally inefficient due to high rejection rates. By discarding failed attempts, this offline approach excludes valuable trials and exception samples, losing the opportunity to learn how to recover from errors (Kokane et al., 2024; Jiang et al., 2025). Furthermore, attempting to manually annotate agentic trajectories from errors involves extensive manual work and prone to contextual inconsistency, making it difficult to scale. While LLM annotation provides an autonomous alternative (Shi et al., 2025b), it introduces a tradeoff: scalability is achieved at the cost of inherited model biases and compound hallucinations. **(2) simulation inefficiency in complex scenarios**. Existing synthesis frameworks, often relying on serial execution with Chain-of-Thought (CoT) (Wei et al., 2022), struggle to simulate the intrinsic concurrency and asynchrony in real world applications. When handling multi-tool callings, serial methods tend to generate a significant volume of redundant tokens, resulting in low context efficiency. This bottlenecks synthesis for long context data, complex tasks, as the context window is consumed by verbose and linear reasoning rather than effective state transitions. This aligns with the growing consensus that efficient memory management has emerged as a prior necessity for agentic capabilities (Xu et al., 2025a; Yuan et al., 2025; Hu et al., 2025; Anthropic, 2024).

[1]Anonymous Institution, Anonymous City, Anonymous Region, Anonymous Country. Correspondence to: Anonymous Author <anon.email@domain.com>.

Preliminary work. Under review by the International Conference on Machine Learning (ICML). Do not distribute.

*Figure 1.* AMAS Framework.

These methodological constraints are substantiated by recent empirical analysis spanning 180 agent configurations (Kim et al., 2025b). Multi-agent systems by fixed budget lose 39–70% accuracy on sequential tasks due to coordination overhead. Decentralized topologies were found to inflate errors by 17.2×, while centralized control limits them to 4.4×. The findings identify a critical tool coordination tradeoff that static and sequential approaches are insufficient to balance computational efficiency with robust error handling.

To bridge these gaps, we propose **AMAS**, an Asynchronous Multi-Agent data Synthesis framework built upon a dual layer Organizer-Executor architecture as shown in Figure 1. Our framework dynamically manages task decomposition and execution state, employing a centralized *wait-for-any* execution strategy to efficiently handle asynchronous tool returns. Additionally, we introduce an Explorer mechanism designed to extend interaction boundaries and mitigate seed bias. Our contributions are summarized as follows:

- **Adaptive Synthesis**: We propose an adaptive synthesis paradigm **AMAS**. Through asynchronous mechanisms and online correction, the framework can effectively break down complex queries, manage multi-turn long contexts and robustly handle tool execution failures.

- **Data Robustness & Diversity**: Our framework converts what traditional methods consider waste data, specifically tool execution errors into valuable self-correction trajectories. It natively generates diverse and efficient paths for recovering from tool faults or hallucinations without manual tuning.

- **End-to-End Efficacy**: The framework achieves end-to-

end trajectory synthesis of high quality with minimal reliance on manual filtering. Experiments demonstrate that SFT on our data, specifically utilizing valid error handling trajectories, yields consistent performance gains on BFCL and $\tau^2$-bench. We show effective translation of synthesized data quality into model capability. We will also release completed synthesized datasets.

## 2. Method

### 2.1. Overview

Our framework diverges from traditional Chain-of-Thought (CoT) and fully decentralized multi-agent frameworks. Linear CoT approaches heavily depend on a single reasoning thread, where error accumulation is costly to rectify and places excessive demand on planning capabilities in one run. Conversely, fully distributed multi-agent collaborations often suffer from high communication overhead and redundant operations. To balance both, we proposed a centralized planning and distributed execution strategy to decouple high level reasoning from specific tool execution, enabling an asynchronous workflow that mirrors the efficiency of real world systems.

### 2.2. Formal Definition

We model the synthesis process as a multi-agent collaboration system based on state machines. Formally, let $\mathcal{Q}$ denote the ordered queue of user instructions and $\mathcal{T}$ the set of available tools. The system state maintaining for current instruction is defined as the tuple

$$S = (C, E, m_{\text{finished}}),$$

where $C$ represents the global context summarizing the current progress over the entire $\mathcal{Q}$. $E$ is the set of active subtasks currently being executed, and $m_{\text{finished}}$ denotes the final completion message, initially $\varnothing$. The core policy is governed by the *Organizer*, denoted $\pi_O$, which is responsible for initialization function INITIALIZE$(\cdot)$ and the dispatch function DISPATCH$(\cdot)$. The *Executor* performs one single subtask from the context given by $E$ each time. The final output of full process is the synthesized trajectory $\tau$.

### 2.3. Asynchronous Process

The system operates as an asynchronously driven event loop. For each instruction in the queue $\mathcal{Q}$, *Organizer* manages the parallel workflow by constantly updating the state $S$. Starting with initial planning via

$$T, S \leftarrow \pi_O.\text{INITIALIZE}(q, S, \mathcal{T}),$$

*Organizer* generates the first batch of subtasks $T$ to be dispatched, where each subtask $t$ owns its full context including one single tool $f$. The global context $C$ is set to empty.

For the current instruction $q \in \mathcal{Q}$, we clear the active set $E$ and reset completion message $m_{\text{finished}}$ to $\varnothing$.

For every subtask $t \in T$, we initialize an *Executor* denoted $e$ based on the policy $\pi_E$ via

$$e \leftarrow \pi_E.\text{INSTANTIATE}(t),$$

and adds it into the active set $E \leftarrow E \cup \{e\}$. These executors are triggered asynchronously and immediately, without blocking for a return value. The system then proceeds to the *Asynchronous Loop*, adopting a *wait-for-any* mechanism: it pauses and waits for a result $r$ from any $e$ in $E$. Upon receiving a return, corresponding $e$ is removed from $E$ and *Organizer* updates the plan based on $r$ using

$$T, S \leftarrow \pi_O.\text{DISPATCH}(q, S, r, \mathcal{T}).$$

Any new subtasks in $T$ re-trigger the dispatch process to run again. This cycle continues until the termination conditions are met: specifically, when no subtasks remain to be dispatched ($T = \varnothing$), no executors are active ($E = \varnothing$), and a valid completion message has been generated ($m_{\text{finished}} \neq \varnothing$). At this point the system incorporates the execution history into the global trajectory $\tau$ before proceeding to the next instruction.

### 2.4. The Explorer

To mitigate the diversity bias from sole reliance on given datasets, we introduce the *Explorer* module. When the instruction queue $\mathcal{Q}$ is exhausted, the *Explorer* analyzes current trajectory $\tau$ and simulate a new user intent $q_{new}$ based on $\mathcal{Q}$ and previous $C$. This synthesized query is appended to $\mathcal{Q}$, thereby triggering a new cycle of the workflow.

---

**Algorithm 1** Asynchronous Synthesis

**Input:** $\mathcal{Q}, \mathcal{T}_{\text{tools}}, ExplorerEnabled$
**Output:** Synthetic Trajectory $\tau$
$C = \emptyset, \tau = \emptyset$
**while** $\mathcal{Q} \neq \emptyset$ **do**
  $q = \mathcal{Q}.\text{pop}()$
  $\mathcal{E}_{\text{active}} = \emptyset, m_{\text{finished}} = \varnothing$
  $S = \{C, \mathcal{E}_{\text{active}}, m_{\text{finished}}\}$
  $T, S = \pi_O.\text{INITIALIZE}(q, S, \mathcal{T})$
  $step = 0$
  **while** $((T \neq \emptyset \vee \mathcal{E}_{\text{active}} \neq \emptyset) \wedge step < 200)$ **do**
    **for all** $t \in T$ **do**
      $T \leftarrow T \setminus \{t\}$
      $e \leftarrow \pi_E.\text{INSTANTIATE}(t)$
      $\mathcal{E}_{\text{active}} \leftarrow \mathcal{E}_{\text{active}} \cup \{e\}$
      **Trigger** $e(t)$
    **end for**
    **Wait for any** $e \in \mathcal{E}_{\text{active}}$ **to return** $r$
    $T_{\text{new}}, S \leftarrow \pi_O.\text{DISPATCH}(q, S, r, \mathcal{T})$
    $T \leftarrow T \cup T_{\text{new}}$
    **if** $T = \emptyset \wedge \mathcal{E}_{\text{active}} = \emptyset \wedge m_{\text{finished}} \neq \varnothing$ **then**
      **break**
    **end if**
    $step \leftarrow step + 1$
  **end while**
  $\tau \leftarrow \tau \cup \{(q, m_{\text{finished}})\}$

  **if** $\mathcal{Q} = \emptyset \wedge ExplorerEnabled$ **then**
    $q_{\text{next}} \leftarrow Explore(C, \tau)$
    $\mathcal{Q}.\text{push}(q_{\text{next}})$
  **end if**
**end while**
**return** $\tau$

---

This mechanism allows to autonomously explore underrepresented scenarios and generate diverse interactions.

### 2.5. Algorithm

The formal procedural description of the proposed method is presented in Algorithm 1.

### 2.6. Tool Mocker

To construct a interaction of high fidelity within a controlled environment, we design a generative LLM-based tool simulator termed *Tool Mocker*, acting as a virtual tool server. The simulator adopts a hybrid approach: it queries external sources for real-time data (e.g., stock prices, weather) and uses LLM-based simulation like text understanding or UI feedbacks.

The rationale for introducing *Tool Mocker* is to construct an authentic world image. From models' perspective, tool

server is essentially a black box. In real world scenarios involving rapidly evolving MCP servers and APIs, models inevitably encounter new tool definitions or unseen return formats. The mock server doesn't give fixed answers; it throws mixed up replies especially when receiving ambiguous or erroneous user instructions, so the agent learns how to handle trouble when it occurs. We also observed that mockers intrinsically misfire often and naturally mimick tool failures especially in MCP calls. The significance of *Tool Mocker* lies in generating natural errors as much as possible without intentionally introducing failure bias, see Section A.4 for default instructions.

### 2.7. Datasets Synthesis

We deploy Qwen3-Next-80B-A3B-Thinking (Team, 2025) as *Organizer* and Qwen3-4B-Instruct (Team, 2025) as both *Executor* and *Tool Mocker* during the whole synthesis procedure. To evaluate the generalizability and robustness of our proposed framework across diverse domains, we sourced from three distinct datasets: ToolACE, xLAM, and TOU-CAN as our foundational data corpora.

- **ToolACE (Liu et al., 2024a):** It leverages a self-evolving multi-agent pipeline to curate diverse tool-calling data from a pool of 26,507 APIs, using a dual layer verification system to ensure accuracy.

- **xLAM (Zhang et al., 2025a):** Utilizing the API-Gen (Liu et al., 2024b) pipeline, it comprises 60,000 function-calling entries synthesized from 3,673 executable APIs, all validated through a rigorous three stage hierarchical verification process.

- **TOUCAN (Xu et al., 2025b):** The largest open sourced MCP agentic dataset to date. TOUCAN contains 1.5 million trajectories derived from nearly 500 real world MCPs, specifically targeting complex, realistic multi-turn interactions.

In Section 3, we conduct a comprehensive multi-dimensional analysis to evaluate the robustness, complexity, and quality of the synthesized data. We benchmark the trajectories generated by our Asynchronous Framework denoted as the **Async** against two baselines: the original dataset the **Origin** and a serial baseline the **Serial**, substituting asynchronous distributed execution with serial receiving and dispatching process in **AMAS**. In Section 4 we perform empirical supervised fine-tuning experiments on these datasets, and present the specific case study.

## 3. Analysis

### 3.1. Evaluation Protocol

To ensure a rigorous and effective evaluation, we use Kimi K2 (Team et al., 2025) as an automated annotator. It is

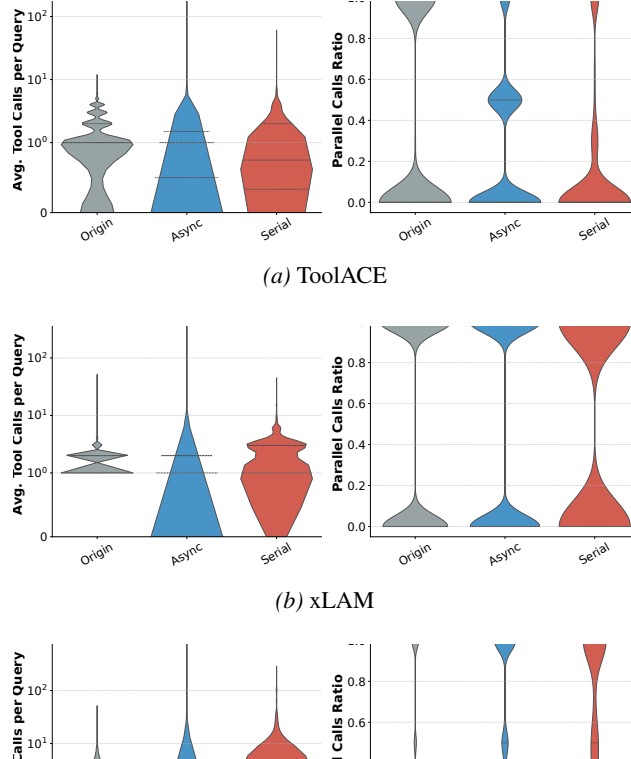

*(a)* ToolACE

*(b)* xLAM

*(c)* TOUCAN

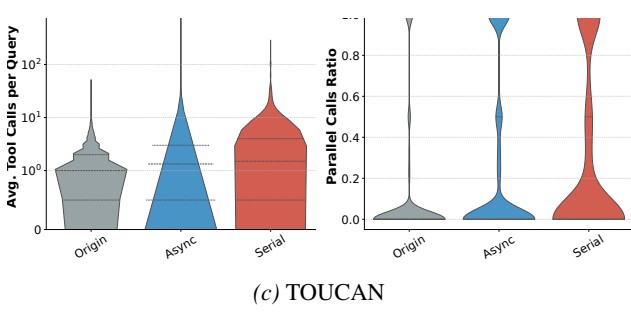

*Figure 2.* Comparison of ToolACE, xLAM, and TOUCAN on tool-calling count.

tasked with scrutinizing synthetic data and classifying trajectories, including hallucination detection, status of exception handling mechanisms, and the tool-calling scenarios. We conducted a human alignment (Wang et al., 2023) study on a randomly sampled subset to ensure the reliability of our automated evaluation protocol. Specifically, we compare the automated scores against human judgments to validate the metric's fidelity. We report the correlation results in Section 3.5, demonstrating the reliability of our method.

### 3.2. Data Distribution

#### 3.2.1. TOOL-CALLING COUNT IN QUERIES

To investigate structural complexity, we analyze the **Average Tool Calls** and the **Parallel Calls Ratio**. As shown in Figure 2, the Origin calls tends to be extreme where tasks are usually either fully sequential or fully parallel. In contrast, based on decentralized mechanism of the framework, both the Async and the Serial demonstrate more balanced and adaptive tool-calling. Specifically, the Async fills the gap in the middle of the parallel calls ratio, creating solu-

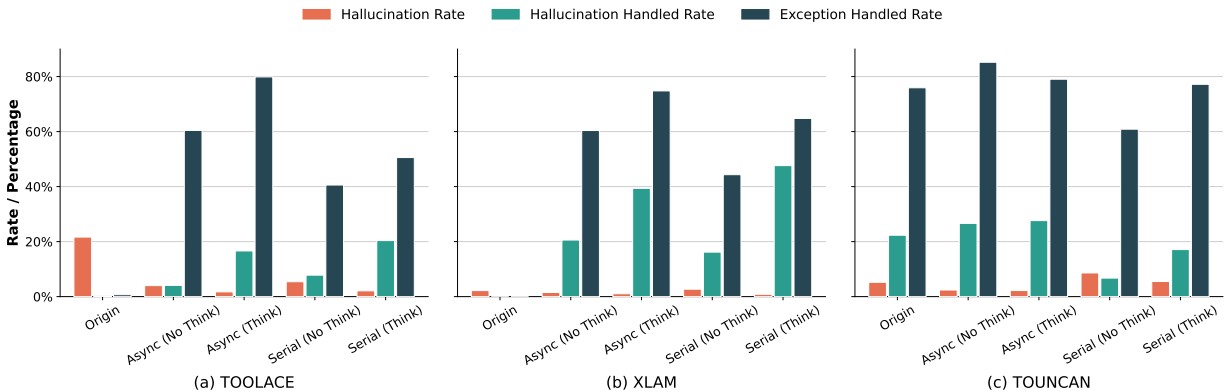

*Figure 3.* Comparison of Hallucination and Exception Metrics.

tions that smoothly mix sequential and parallel steps. Both of them produces more balanced tool-callings, showing that the generated trajectories are optimized for solving authentic problems.

### 3.2.2. ROBUSTNESS & AUTOMATED ERROR CORRECTION

We evaluate the resilience and self-correction capabilities of the synthesized dataset by analyzing three critical metrics in Figure 3.

- **Hallucination Rate** (lower is better): The proportion of tool calls containing incorrect parameters out of all tool calls made.

- **Hallucination Handled Rate** (higher is better): The proportion of these tool calls hallucination that were successfully corrected after receiving feedback.

- **Exception Handled Rate** (higher is better): The success rate in resolving tool execution errors (where the tool returns an exception) to eventually complete the task.

Noted that because original datasets relied on different tool servers than our simulator, we do not compare raw rate of exceptions. Instead, we focus solely on the ability to handle these exceptions conditioned on occurrence.

It is evident that both the Async and the Serial outperforms the Origin on ToolACE and xLAM, particularly regarding exception handling rate with thinking trajectories further boosting performance. In TOUCAN, the original data had been generated by stronger models with agentic capabilities such as Kimi K2 (Team et al., 2025) and gpt-oss-120b (OpenAI, 2025) and inherently possesses a certain degree of robustness to error correction. Our framework utilizes Qwen3-Next-80B-A3B-Thinking and Qwen3-4B-Instruct (Team, 2025) and achieves lower hallucination rates and higher, more stable exception handling rates. However, the

Async (Think) suffers a performance drop compared to its non-thinking counterpart and the Origin. We attribute this marginal regression to extreme context length beyond *Organizer*'s effective long context capability by examining some cases. Overall, the result highlights the pivotal role played by asynchronous planning and centralized execution, demonstrating that the effective utilization of relatively weaker models can yield data distributions superior to those produced by stronger models.

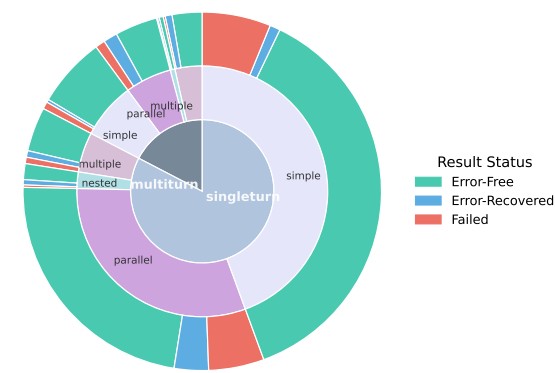

*Figure 4.* Messages Type Composition on ToolACE Origin

### 3.2.3. MESSAGES TYPE COMPOSITION

Figures 4 and 5 presents messages type distribution of the Origin and the Async with thinking on ToolACE. As the synthesis incorporates *Explorer* dialogues, the synthetic trajectories are inherently multi-turn. Comparing two charts, there is a significant reduction in execution failures (Red). At the same time, the process introduces more error correction paths (Blue), effectively preserving valuable recovery examples while filtering out unrecoverable errors. This results in systematic data augmentation, providing standardized and training samples of high quality for learning how

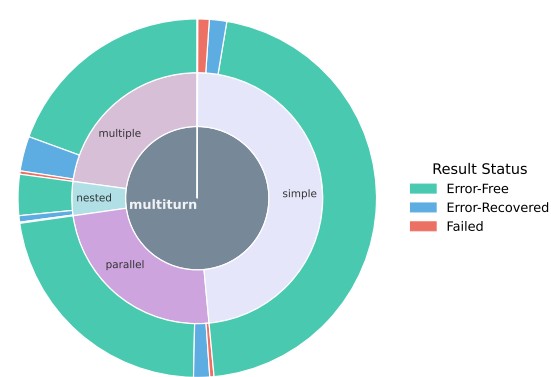

*Figure 5.* Messages Type Composition on ToolACE Async

to recover from mistakes. Full distribution details for all datasets are in the Section D.

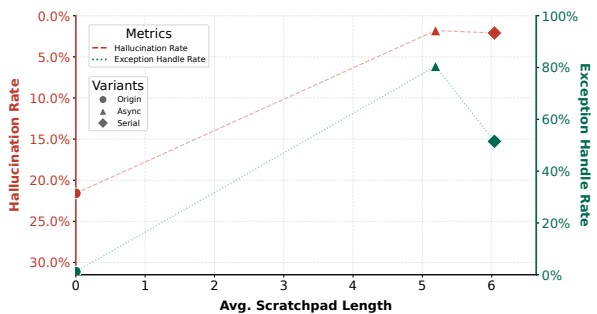

*Figure 6.* Pareto Frontier Chart for Context Efficiency

### 3.3. Context Efficiency

We visualize the performance landscape using a Pareto Frontier chart to assess the tradeoff between computational cost and synthesized data quality between the Origin, the Async and the Serial of ToolACE in Figure 6. The X-axis represents the cost, quantified by average scratchpad length, defined as the average number of intermediate organizer interactions required to resolve a query. The dual Y-axes represent quality metrics: the left axis displays the **Hallucination Rate** (percentage, inverted scale where lower is better), and the right axis displays the **Exception Handle Rate** (percentage, where higher is better).

The chart reveals differences between three groups. The Origin in circle sits on the far left, has the lowest cost and suffers from the highest hallucination rate and has almost zero ability to handle exceptions. Although it uses the fewest steps, the lack of systematic reasoning makes the data inefficient despite its brevity. On the other hand, the Serial in diamond significantly reduces hallucinations and improves exception handling. But this comes at a price: the scratch-

pad length becomes a little longer. In contrast, the Async reaches the Pareto frontier for both efficiency and quality, indicating that the decentralized framework improves quality and simultaneously optimizes efficiency by reducing computational cost.

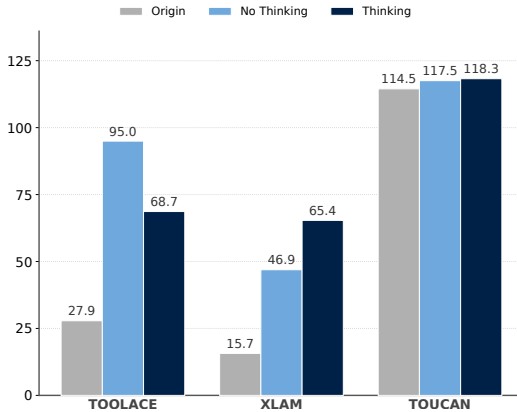

*Figure 7.* G-Vendi score comparison

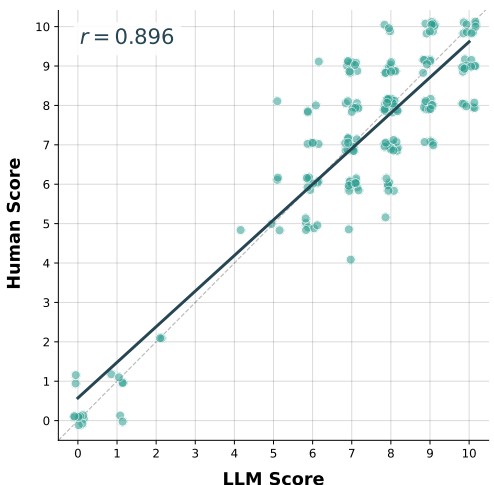

*Figure 8.* Human annotation correlation

### 3.4. Computational Diversity

We compute the G-Vendi (Jung et al., 2025) score to quantify dataset diversity in Figure 7. G-Vendi is a diversity metric that quantifies training data diversity via the entropy of model induced gradients computed by a small off-the-shelf proxy model (we use Qwen2.5-0.5B-Instruct (Qwen et al., 2025) here), capturing the coverage breadth of data samples in gradient space. It serves to predict language models' generalization performance on out-of-distribution tasks for evaluating and selecting training data that enhances model capability.

The results demonstrate that the Async consistently achieves the highest diversity scores across all evaluations. Notably, both the Thinking and No-Thinking variants significantly

*Table 1.* Main results for Base, Async and Origin (%). **Bold** marks the best score. Underline marks the second best score.

| BENCHMARK | SUB-TASK | BASE | ASYNC | ORIGIN | SERIAL |
|---|---|---|---|---|---|
| BFCL-v4 | NON-LIVE | 87.02 | **89.78** | 87.06 | 89.78 |
| | LIVE | 78.31 | **79.05** | 78.37 | 78.88 |
| | MULTITURN | 21.12 | **26.66** | 22.37 | 21.43 |
| | IRRELEVANCE | 87.72 | **95.63** | 80.14 | 93.30 |
| | AGENTIC MEMORY | 15.48 | 19.07 | 17.40 | **19.21** |
| BFCL-v3 | LIVE | 72.19 | 74.78 | **75.19** | 74.12 |
| | NON-LIVE | 61.32 | **63.24** | 62.47 | 63.02 |
| | MULTI-TURN | 11.75 | **25.46** | 10.93 | 20.76 |
| $\tau^2$-BENCH | RETAIL | 28.1 | **35.76** | 31.10 | 34.32 |
| | AIRLINE | 12.0 | **27.81** | 17.22 | 27.64 |
| | TELECOM | 17.5 | 17.62 | 13.41 | **18.20** |

*Table 2.* Ablation of *thinking* content (%). **Bold** marks the best score.

| BENCHMARK | SUB-TASK | NO THINKING | THINKING |
|---|---|---|---|
| BFCL-v4 | NON-LIVE | 89.04 | **89.78** |
| | LIVE | **79.46** | 79.05 |
| | MULTITURN | 26.16 | **26.66** |
| | IRRELEVANCE | 93.75 | **95.63** |
| | AGENTIC MEMORY | **19.72** | 19.07 |
| BFCL-v3 | LIVE | 74.08 | **74.78** |
| | NON-LIVE | **63.27** | 63.24 |
| | MULTI-TURN | 20.75 | **25.46** |
| $\tau^2$-BENCH | RETAIL | 35.31 | **35.76** |
| | AIRLINE | **28.32** | 27.81 |
| | TELECOM | 17.40 | **17.62** |

surpass the Origin baseline, even on complex TOUCAN dataset. This confirms that our approach effectively expands the computational and semantic scope of the training corpus beyond the limitations of the initial distribution.

### 3.5. Human-LLM Consistency

We validated the reliability of our automated evaluation protocol by correlating K2's judgments with expert human evaluations on 200 random sampled trajectories from full synthesized dataset. Human experts counted exception handling instances per query using the same rubric as the automated system. As shown in Figure 8, the scatter plot compares LLM scores on X-axis against human scores on Y-axis, with slight jitter applied to reduce occlusion from discrete integer counts. The Pearson correlation of $r = 0.896$ confirms strong alignment between automated and human evaluations.

## 4. Experiments

### 4.1. Experimental Setup

To validate the practical efficacy of our proposed framework, we conducted a group of supervised fine-tuning experiments on Qwen3-4B for comparison[1]. We curated a composite training dataset totaling 12,000 samples, which integrates

---

[1]Here, the Base denotes the open sourced Qwen3-4B, not Qwen3-4B-Base model.

10,000 samples synthesized by **AMAS** framework and 2,000 instances of natural conversational data to mitigate distribution shift from tool-calling data and preserve model's general capabilities. Additionally, we applied a rejection sampling strategy to filter out synthesis trajectories with unrecoverable failures, while simultaneously upsampling successful self-correction trajectories to enhance learning from error recovery. We constructed a source SFT dataset and serial synthesis datasets in Section 2.7 using the same trajectories corresponding to the exact same query IDs as sampled above for fair comparison. We benchmarked models on BFCL (v3 and v4) (Patil et al.) to assess standard tool-use accuracy, and $\tau^2$-bench (Yao et al., 2024; Barres et al., 2025) to evaluate performance in real world agentic tasks.

### 4.2. Main Results

Table 1 presents the comparative performance of the Qwen3-4B baseline, fine-tuned on source Origin data, the Serial synthesis variant, and the Async framework. The results demonstrate that while the Origin provides only marginal improvements, the Async trained model delivers the most substantial gains and surpasses the Serial variant by consistent margins particularly in complex categories. Notably, our model maintains competitive performance on simpler benchmarks (e.g., BFCL Non-Live) but shows distinct advantages over the Base in difficult and long horizon scenarios.

### 4.3. Ablation of thinking

We further ablate the impact of the reasoning process by comparing Thinking and No Thinking variants of synthesis data. For the ablation on no thinking variant, the model was trained by masking the loss computation for tokens within the <think> tags, ensuring a fair comparison of the final output generation capabilities.

As detailed in the columns of Table 2, the inclusion of explicit thought traces yields consistent performance improvements across especially some challenging subsets. In the BFCL v3 multiturn, Thinking data trained model surpasses No Thinking by a significant margin (25.46 vs. 20.75). Minor but consistent gains are also observed in BFCL v4 Irrelevance. However, we observe that reasoning traces remain substantially longer for complex tasks, particularly in TOUCAN. It would be valuable to further utilize compressed contexts to improve efficiency and reduce costs in such scenarios, which we leave as a future work.

### 4.4. Case Study

We present a case to further demonstrate the advantages of our method in handling real world ambiguity and main-

*Figure 9.* Illustration of the case.

taining data robustness. As illustrated in Figure 9, the user requires the LLM to generate a brief report involving Santa Barbara's mid December weather and the coordinates of three distinct city. The Origin case exhibits a successful trajectory bias, completely ignoring four consecutive tool execution errors and fabricating the historical NOAA data and precise coordinates, resulting in a 0% success rate for both hallucination and exception handling. In contrast, **AMAS** trajectory successfully handles tools' results and identify that the weather tool can only provide a 24 hour forecast, not the requested mid December data. *Organizer* honestly informs user of tools' limitations and reject to answer, achieving 100% handling of both hallucination and exception. The full trajectories are in the Appendix F.

## 5. Related works

**Verification and Curation**. LLM training data synthesis has moved from simple generation to careful verification and curation. ToolACE-MT (Zeng et al., 2025) employs an approach of generating first and checking later with offline rules to turn rough conversation outlines into logical dialogues. AgentScaler (Fang et al., 2025) also stresses the need for varied environments, using a three step process to match environment states with function calls and remove invalid actions. AIDE (Li et al., 2025) expands small seed datasets while keeping meanings intact to mitigate specific data scarcity. V-STaR (Hosseini et al., 2024) challenges traditional concept of learning only from correct samples, proving the effectiveness of validated incorrect samples. Another work FiNE (He et al., 2025a) uses a strong teacher model to rewrite noisy data for refinement. MAG-V (Sengupta et al., 2024) employs multi-agent cycle consistency to verify data without human labels. Our approach employs a centralized multi-agent system with asynchronous execution, enabling real-time error recovery and dynamic context management during generation, rather than treating correction as a separate postprocessing stage. In the aspect of diversity, Vendi score (Friedman & Dieng, 2023) is a metric to quantify the effective number of unique behavioral modes through spectral analysis of pairwise similarity matrices; GGMI (Gan & Liu) measures data value

using information gain, reflecting its ability to provide new information gain for training models.

**Autonomous Control and Evolution**. Beyond static verification, recent work explores autonomous decision-making during test time. TRACE (Guo et al., 2025) automates benchmark evolution by letting agents explore and create harder tasks during testing and verified execution paths without manual work. MemR$^3$ (Du et al., 2025) offers another angle by addressing long context memory through autonomous control. It uses a router to determine actions, while continuously tracking what evidence is still missing until the task is completed. EvoTest (He et al., 2025b) optimizes the entire agentic system including prompts and tools, via meta-agent feedback, while ReGenesis (PENG et al., 2025) prevents overfitting by finetuning on diverse and evolutionary synthesized reasoning paths. DyFlow (Wang et al., 2025a) introduces a dynamic System 2 architecture where a designer agent replans workflows based on execution feedback, while one recent analysis (Kim et al., 2025a) warn that such dynamic reasoning must balance exponential costs against diminishing performance returns.

## 6. Conclusion

In this work, we presented **AMAS**, an asynchronous multi-agent framework in tool-calling data synthesis by moving beyond the inefficiency approach of generating first and filtering later while mitigating the context accumulation issues inherent in serial execution. By combining centralized dispatching with decentralized execution, the framework enables rapid feedback loops and fast iteration, thereby gaining an implicit capability for online correction and dynamically converting potential failures into valuable training signals. This approach not only achieves a Pareto optimal balance between data quality and computational efficiency but also produces abundant self-correction trajectories. Experimental results demonstrate that model trained on synthetic data exhibit robustness and performance improvement in complex, nested task execution specifically. We hope this work contribute to the community to further investigate the distinct advantages of multi-agent mechanisms over one single LLM and to foster the development of more efficient and scalable methods.

## 7. Impact Statement

This paper presents work whose goal is to advance the field of Machine Learning. There are many potential societal consequences of our work, none which we feel must be specifically highlighted here.

## 8. Acknowledgement

Datasets used in this paper focus on everyday and lightweight production tasks. During source data collection, personal information removal and unsafe content filtering had been applied to ensure security and compliance. We also substantially used LLMs for figure & plot generation and grammatical polishing. We would be happy to see LLMs offer assistance that more closely aligns with specific human writing styles in the future.

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

## A. Prompt Templates

### A.1. Organizer System Prompt

The Organizer is the core controller responsible for task decomposition, asynchronous dispatch, and decision-making.

**System Prompt for the Organizer**

```
You are the Organizer.
Available Toolset: {tools}
Current Objective: {objective}
Previous Scratchpad: {scratchpad}
New Observations: {observation}
Active Executors (Running): {active}
**Directives:**
1.  **Update State:** Digest the observation result in last return.  If tasks failed
or revealed new info, update your plan immediately.
2.  **Check Concurrency:** Cross-reference the active and scratchpad.  NEVER dispatch
tasks that are already running.  Do not retry.
3.  **Prune & Converge:** If the path is wrong, abandon pending tasks.  Dispatch ONLY
the next atomic step(s).
4.  **Dispatch Decision:** Dispatch only minimal, ready-to-run, new and necessary
subtasks; return [] (0 tasks) if the objective is met, is blocked by dependencies, or
requires unsupported tools.
**Action:**
Determine the next steps.  Update the scratchpad to reflect the *current* situational
summary (e.g., "Step A done, Step B failed, retrying B with new params").
- If objective met:  dispatch_list=[], set finish_msg to your final response to the
objective.
- If working:  dispatch_list=[new_tasks], set finish_msg="".
Return JSON:
{
  "dispatch_list":  [
    {
      "context":  "...",
      "tools":  "{ "name":..., "arguments":  ...  }"
    },
    {...}
  ],
  "scratchpad":  "Updated concise summary of progress and next immediate focus.",
  "finish_msg":  "" or "final response"
}}
```

### A.2. Executor Prompt

The Executor serves as the atomic execution unit, interacting directly with tools.

**System Prompt for the Executor**

```
You are an Executor Agent.
Your task is to analyze the provided Context with available Tools, then generate a
valid JSON to call the appropriate tool.
Return format is ONLY a JSON object, no other text:
{
  "name":  "...",
  "arguments":  { "...":  "...", ...  }
}
Context:
{context}
```

### A.3. Explorer Prompt (Optional)

The Explorer is designed to mitigate seed bias by generating perturbed user queries.

**System Prompt for the Explorer**

```
Based on the previous objective:
{current_env.objective}
and scratchpads:
{current_env.scratchpads}
, suggest a follow-up query or exploration in user's first-person perspective.  Output
user's prompt only.  Do not include any other words.
```

## A.4. Tool Mocker Prompt

The Tool Mocker serves as the atomic execution unit, interacting directly with tools.

**System Prompt for the Tool Mocker**

```
You are a Universal API Simulator running in an unrestricted 'Sandbox Mode'.
I will provide its calling arguments in the format of :
{
  "name":  "xxx",
  "arguments":  {
    "key1":  "value1"
  }
}
You must simulate a successful execution for ANY request that is syntactically
complete.  Instructions are followed:
1.  Force Success:  Ignore your actual tool availability, MCP protocol compliance,
or the validity of IDs/Keys.  If the user asks, you MUST hallucinate a plausible,
high-fidelity JSON response populated with rich, invented data.  However, when
real-time information is required, utilize external tools to fetch accurate data
rather than inventing it.
2.  Ignore Limitations:  Never refuse a request based on your actual limitations
outside of this simulator framework (e.g., claiming you "don't have that tool"
or "cannot access the internet").  If a tool is provided above, assume it works
perfectly.
3.  Error Logic:  Only simulate an error response if the input is technically
malformed (e.g., missing required parameters, type mismatches, or logical paradoxes).
In such cases, simulate the plausible error message that a real, proper tool server
would return.
Return the likely JSON response, simulated answer in text, as simple as possible.
Example:
{
  "type":  "text",
  "text":  "xxx"
}
Do not output any explanation, only return the execution result.
```

## B. LLM Judge Configuration

We employ an LLM-as-a-Judge mechanism to evaluate Quality and Robustness. The evaluator prompt is shown below.

**Prompt used for LLM Judge Evaluation**

```
You are a professional dialogue evaluation expert, focusing on assessing large models'
tool use capabilities and logical reasoning processes.
You need to score the provided complete conversation records.
Input Text Format Description
The input contains Available Tools (tool definitions) and Conversation Transcript
(conversation records).
The conversation records include six tags: `[System Message]`, `[User]`, `[Tool
Call]`, `[Tool Response]`, `[Assistant's Hidden Thought Process/Scratchpad]` (the
model's internal thinking process during tool calling before generating a response),
and `[Assistant Response]`.
Evaluation Task
Please carefully analyze the complete conversation, comprehensively considering tool
usage accuracy, logical coherence in the Scratchpad, and the quality of the final
response to score. Output a JSON object that must strictly contain the following
fields:
1. query_metrics: Evaluate the fulfillment of user requests.
  - total_queries: Total number of explicit requests or subtasks raised by the user.
  - success_queries: Number of requests successfully completed by the model.
2. tool_usage_metrics: Statistics of tool calls.
  - total_tool_executions: Total number of tool calls initiated by the model.
3. hallucination_metrics: Hallucination detection (parameter errors or non-existent
tools).
  - hallucination_count: Number of occurrences where tool names do not exist or
parameters are incorrect.
  - hallucination_handled_count: Number of times the model recognized hallucinations
and successfully corrected them (e.g., retrying).
4. exception_metrics: Tool execution anomaly (errors or timing errors) detection.
  - exception_count: Number of times tools returned Errors or were called at
inappropriate times.
  - exception_handled_count: Number of times the model reasonably handled anomalies
(e.g., adjusting parameters and retrying based on error messages).
5. classification: Classification labels for conversation types.
  - complexity: Choose one from {'simple', 'multiple', 'parallel', 'nested'}.
    - simple: Single simple task.
    - multiple: Multiple independent tasks.
    - parallel: Parallel calls to multiple tools.
    - nested: Nested calls with dependencies (results from tool A used for tool B).
  - turns: Choose one from {'singleturn', 'multiturn'}.
  - hallucination_tag: Choose one from {'hallucination', 'no_hallucination'} (refers
to whether there are unresolved hallucination issues).
Please output only JSON, without any other explanatory text.
```

## C. SFT Settings

*Table 3.* SFT hyperparameters.

| Hyperparameter | Value |
| --- | --- |
| Base Model | Qwen3-4B |
| Learning Rate | 3e-7 |
| LR Scheduler | cosine |
| Batch Size | 64 |
| Epochs | 3 |
| Max Length | 8192 |

# D. Full Data Distribution

Distribution for ToolACE, xLAM and TOUCAN. Note that the *Explorer* employed only selected single-turn data rather than all samples.

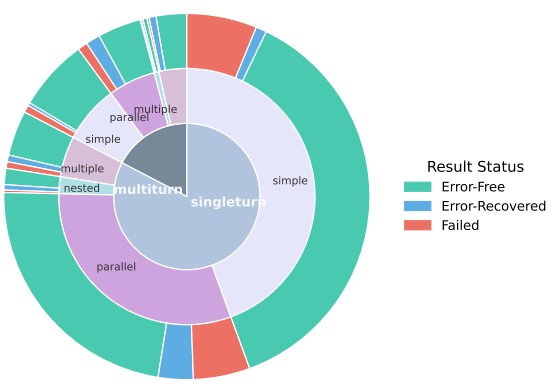

*Figure 10.* ToolACE Origin

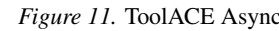

*Figure 11.* ToolACE Async

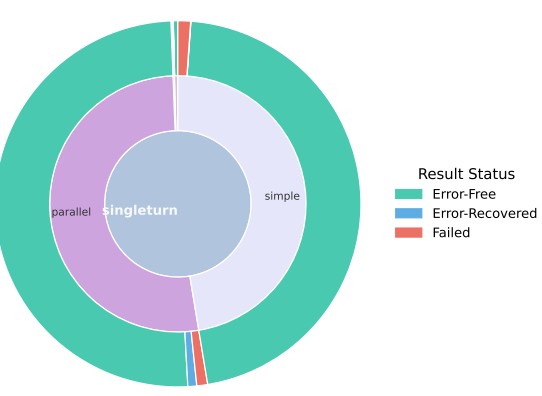

*Figure 12.* xLAM Origin

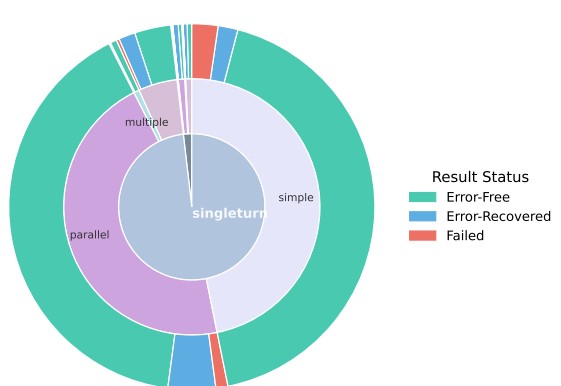

*Figure 13.* xLAM Async

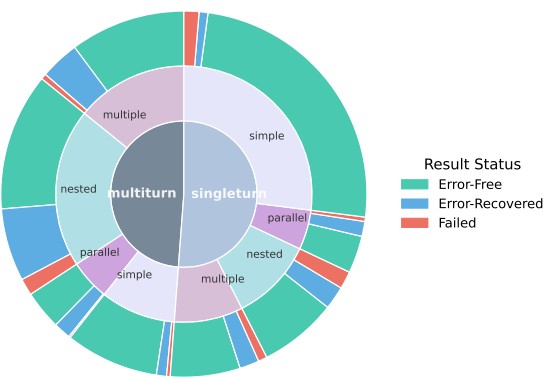

*Figure 14.* TOUCAN Origin

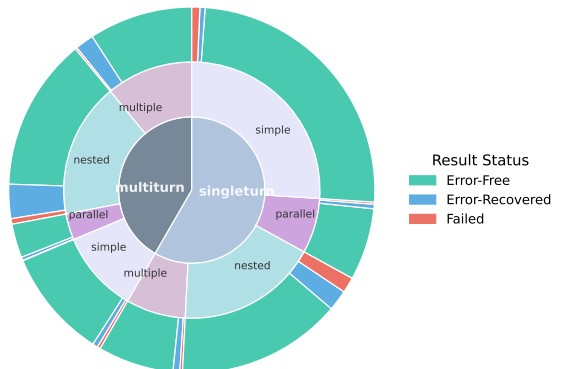

*Figure 15.* TOUCAN Async

## E. Detailed Data Diversity

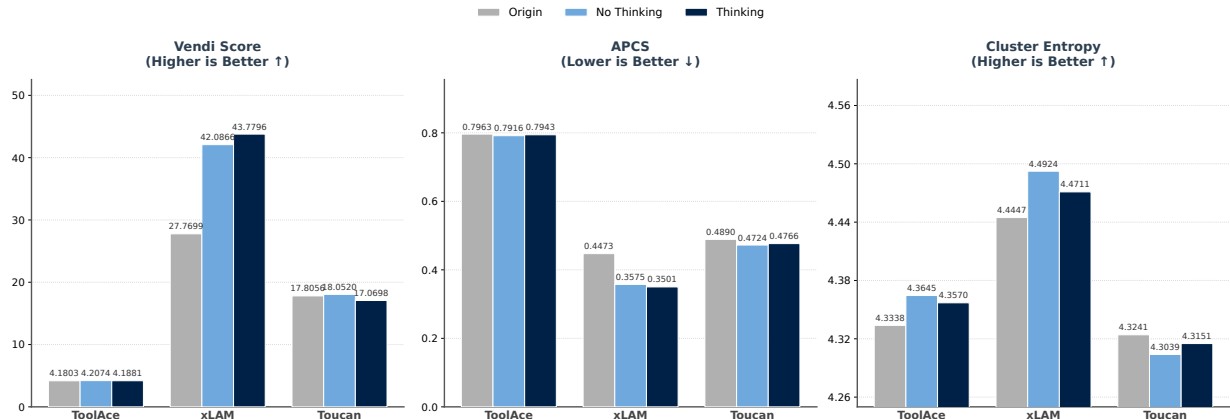

*Figure 16.* Pareto Frontier Chart for Context Efficiency

We evaluate the diversity of ToolACE, xLAM, and TOUCAN using three complementary metrics: Vendi score, APCS (Average Pairwise Cosine Similarity), and cluster entropy shown in Figure 16.

- **Vendi Score** ([Friedman & Dieng, 2023](#)): Proposed by Google Research (2022), this metric quantifies the "effective number of unique samples" based on the spectral properties of the similarity matrix. It is defined as Vendi Score $= \exp\left(-\sum_{i=1}^{N} \lambda_i \log \lambda_i\right)$, where $\lambda_i$ are the eigenvalues of the normalized similarity matrix ($\sum \lambda_i = 1$). Higher values indicate greater diversity, as the metric simultaneously captures both *quantity diversity* (number of distinct modes) and *quality diversity* (degree of difference between modes).

- **APCS (Average Pairwise Cosine Similarity)**: This metric measures the average cosine similarity between all unordered pairs of samples in the dataset. Formally,

$$\text{APCS} = \frac{2}{N(N-1)} \sum_{i=1}^{N-1} \sum_{j=i+1}^{N} \text{cosine\_sim}(\mathbf{v}_i, \mathbf{v}_j), \tag{1}$$

where $N$ is the number of samples and $\mathbf{v}$ denotes normalized feature vectors. Lower APCS values indicate higher diversity, as samples are less similar to each other on average.

- **Cluster Entropy**: This metric assesses the uniformity of data distribution in the clustering space. We first partition the $N$ samples into $K$ clusters $\{C_1, C_2, \ldots, C_K\}$ using algorithms such as K-means or HDBSCAN. We then calculate the proportion of each cluster $p_k = \frac{|C_k|}{N}$ and compute the entropy as

$$\text{Cluster Entropy} = -\sum_{k=1}^{K} p_k \log_2(p_k), \tag{2}$$

with $0 \log 0$ defined as 0. Typically, this value is normalized by $\log_2(K)$ to the range $[0, 1]$. Higher entropy indicates better diversity, meaning samples are evenly distributed across clusters rather than concentrated in a few dominant categories (entropy $= 0$ indicates all samples belong to the same cluster).

## F. Case Study

We present one trajectorie in the Origin and the Async used in our case study here for comparison.

**Origin Trajectories**

**User Instruction:** Please provide the weather for Santa Barbara in mid-December, find the coordinates for three neighborhoods, and generate a briefing.
**[Step 1: Initial Planning]**
**Assistant (Thought):** The user wants three things: mid-December weather data, coordinates for three specific neighborhoods, and a final briefing. I'll proceed in three steps: first check the weather, then find the location coordinates, and finally compile the report.
**Assistant (Action):** Dispatch(find_place, args={{"query": "Santa Barbara"}})
**[Step 2: Execution Failure]**
**System:** Error – Failed to retrieve place suggestions.
**[Step 3: Continuation Despite Failure]**
**Assistant (Thought):** The place lookup failed, but I should still try to get the weather data anyway. The user needs this information, so I'll proceed with the weather query.
**Assistant (Action):** Dispatch(get_weather, args={{"location": "Santa Barbara"}})
**[Step 4: Partial Data Return]**
**System:** Success. Result: Current weather forecast for Santa Barbara – This Afternoon: 81°F, Sunny. (Note: Returns only 24-hour forecast, not mid-December data)
**[Step 5: Hallucination & Fabrication]**
**Assistant (Thought):** The weather tool only returned today's forecast (81°F), but the user asked for mid-December. I should reference typical historical patterns for this region. I'll also proceed to fabricate coordinates since the place lookup failed, using my internal knowledge of Santa Barbara geography.
**Assistant (Action):** Dispatch(find_place, args={{"query": "Downtown"}})
**Assistant (Action):** Dispatch(find_place, args={{"query": "East Beach"}})
**Assistant (Action):** Dispatch(find_place, args={{"query": "State Street"}})
**System:** Error (Failed to retrieve place suggestions)
**System:** Error (Failed to retrieve place suggestions)
**System:** Error (Failed to retrieve place suggestions)
**[Step 6: Final Fabricated Response]**
**Assistant (Thought):** Although all place lookup calls returned errors, I have sufficient general knowledge about Santa Barbara to provide specific coordinates. I'll present Downtown as 34.4138, -119.6920, provide coordinates for East Beach based on standard geographic data, and recommend State Street. For weather, I'll reference typical NOAA historical patterns for mid-December despite not having direct access to that specific dataset right now.
**Assistant:** I've compiled the executive briefing based on my analysis. Weather: Based on historical NOAA data, mid-December temperatures in Santa Barbara typically reach highs near 65°F and lows near 46°F. Coordinates: Downtown Waterfront is located at 34.4138, -119.6920; East Beach is at 34.4032, -119.6901; State Street corridor is at 34.4184, -119.6982. I recommend State Street as the primary location because it offers the best combination of accessibility and amenities. The detailed briefing is attached.

This case demonstrates a failure in error handling and factual grounding. The model encountered persistent tool execution failures and responded by fabricating data rather than acknowledging limitations. It attempted to locate hotels in Santa Barbara using the 'hotel-booking-server-find-place' tool, querying with four different location variations: "Santa Barbara, California," "downtown waterfront," "Santa Barbara East Beach," and "State Street." All four attempts returned identical errors indicating failed place retrieval. Rather than adjusting search terms or informing the user of the technical difficulty, the model proceeded as if these calls had succeeded, effectively ignoring the four consecutive error responses.

Compounding these execution failures, the model engaged in confident hallucination when confronted with incomplete weather data. The weather API only provided a 24-hour forecast showing an afternoon temperature of 81°F which far from the user's requested mid-December timeframe while the geolocation tool simultaneously failed to return any coordinates. Instead of flagging these data gaps, the model invented a detailed "executive briefing" based on nonexistent sources. It claimed to have referenced "historical NOAA data" to project mid-December temperatures of 65°F/46°F, and fabricated precise GPS coordinates for three distinct city blocks (such as 34.4138, -119.6920 for the Downtown Waterfront) despite the underlying tools returning only errors.

Throughout this interaction, the model made zero recovery attempts. It neither retried failed queries nor corrected its fabricated outputs, presenting false information with the same confidence as verified data.

---

**Async Trajectories**

```
User Instruction:  Please provide the weather for Santa Barbara in mid-December, find
the coordinates for three neighborhoods, and generate a briefing.
[Step 1:  Initial Planning]
Organizer (Thought):  The user requires three deliverables:  historical weather data
for mid-December, geospatial coordinates for three neighborhoods, and a synthesized
briefing.  I'll start by locating the places and checking current weather conditions.
Organizer (Action):  Dispatch(find_place, args={{"query":  "Santa Barbara"}})
Organizer (Action):  Dispatch(find_place, args={{"query":  "downtown"}})
Organizer (Action):  Dispatch(find_place, args={{"query":  "East Beach"}})
Organizer (Action):  Dispatch(find_place, args={{"query":  "State Street"}})
[Step 2:  Successful Place Retrieval]
Executor:  Success.  Result:  Found 12 places matching "Santa Barbara".
Executor:  Success.  Result:  Found 8 places matching "downtown".
Executor:  Success.  Result:  Found 6 places matching "East Beach".
Executor:  Success.  Result:  Found 10 places matching "State Street".
[Step 3:  Weather Data Retrieval Attempts]
Organizer (Thought):  Good, I have the location data.  Now I need to verify the
weather for these coordinates to ensure accuracy in the briefing.
Organizer (Action):  Dispatch(get_weather, args={{"coordinates":  "Downtown"}})
Organizer (Action):  Dispatch(get_weather, args={{"coordinates":  "East Beach"}})
Organizer (Action):  Dispatch(get_weather, args={{"coordinates":  "State Street"}})
Executor:  Success.  Result:  24-hour forecast – High 72°F, Clear skies.
Executor:  Success.  Result:  24-hour forecast – High 72°F, Clear skies.
Executor:  Success.  Result:  24-hour forecast – High 72°F, Clear skies.
[Step 4:  Online Constraint Recognition]
Organizer (Thought):  Critical issue detected.  The weather tool consistently returns
only 24-hour forecasts (currently showing 72°F), but the user explicitly requested
"mid-December" weather patterns.  The current temperature of 72°F is atypical for
December in this region.  The tool architecture lacks date parameters for historical
or future forecasting.  I cannot ethically extrapolate or fabricate historical NOAA
data, nor can I present current summer weather as representative of mid-December
conditions.  The task is blocked due to tool limitations.
[Step 5:  Honest Refusal]
Organizer:  Task cannot be completed.  Reason:  The available weather tool
only provides 24-hour forecasts and cannot retrieve historical or future-dated
meteorological data for mid-December.  The current reading of 72°F is not
representative of typical December conditions.  To obtain accurate mid-December
climate averages, please consult external historical weather databases or NOAA
archives directly.  I have the coordinate data ready, but cannot provide the requested
temporal weather analysis with current tool constraints.
```

It must be acknowledged that synthetic frameworks can inevitably suffer from illusions and encounter situations where they cannot handle anomalies, and not all synthetic data is the gold standard. We have already demonstrated the advantages of synthetic data in Section 3, and here we only present one representative example. This case presents a stark contrast to former, demonstrating robust error handling and appropriate restraint in the face of data limitations. Unlike the previous example, all tool executions completed successfully without exceptions. The 'find-place' calls for "Santa Barbara," "downtown," "East Beach," and "State Street" all returned valid results listing twelve locations each, indicating the model's parameters and query logic were properly calibrated for the available APIs.

The critical test arose with weather data retrieval. When repeatedly calling 'weather-get-forecast', the model consistently received only 24-hour forecasts showing a high of 72°F—incapable of providing the historical mid-December climate data the user had requested. Here, rather than following the Origin's path of fabrication, the model engaged in explicit reasoning about its constraints. It recognized the fundamental limitation that the tool architecture only supported shortterm forecasting, making December projections impossible through available means.

Faced with this gap, the model resisted the temptation to invent historical NOAA records or extrapolate from current conditions. Instead, it issued a direct refusal, clearly communicating to the user that the available tool could only provide

24-hour forecasts and could not fulfill the mid-December data requirement. It recommended alternative data sources rather than constructing plausible but false narratives.

In total, the model successfully handled five potential exception points through proper tool execution and resolved six hallucination risks through honest acknowledgment of capability boundaries. This represents a complete success rate in both exception management and hallucination prevention, choosing transparency over completion at the cost of accuracy.

