# OpenReview forum: "An Asynchronous Multi-Agent Framework for Adaptive Tool-calling Data Synthesis"
_ICML.cc/2026/Conference — Submitted to ICML 2026_

### Official Review · Reviewer_1LcP · 2026-03-13

**Soundness:** 3
**Presentation:** 2
**Significance:** 3
**Originality:** 2
**Overall Recommendation:** 3
**Confidence:** 4

**Summary:**

This paper presents AMAS, a multi-agent data synthesis framework which decouples planning from execution. The framework also includes several additional components to enhance the data quality. The Explorer proposes new data to be generated based on the existing trajectories while a Tool Mocker which acts as a virtual tool server. They analyze the distribution of the data generated by the AMAS technique and use it to finetune a Qwen3-4B model and find that it performs favorably compared to the baselines.

**Compliance With Llm Reviewing Policy:**

Affirmed.

**Final Justification:**

I have read the author's rebuttal, and their rebuttal to the other authors, and I will raise my score to a 3.

Indeed, the paper has very impressive analysis and breakdown of the data generated by AMAS, but my main concern is that Section 2: Method is rather vague across the board. Looking at the response that the authors referred me to in rebuttal Gy3i, it doesn't fully resolve my question about the AMAS framework, as I don't think that the paper clearly documents their method enough to the extent that one could reproduce the results of the paper.

I think the high-level architecture has a very good amount of analysis, but some of these smaller, yet critical, bespoke components such as the Tool Mocker and the Explorer are underdeveloped and not well ablated in the original work. It seems like in rebuttal Gy3i, the Explorer is ablated, but not the Tool Mocker.

However, it seems like through the rebuttal, several components do have ablations attached to them that will be added to the paper, and some of my questions have been cleared up, so I am willing to raise my scores, but I still have significant concerns about the writing of the paper.

**Key Questions For Authors:**

1. Can the authors better explain the unique components of their data synthesis pipeline? This would vastly improve the soundness and presentation of the paper overall.
2. Can the authors run any of the additional ablations detailed above, so that we can see how these different components affect downstream performance.

**Limitations:**

yes

**Strengths And Weaknesses:**

Strengths:
* Section 3, the analysis of the data distribution, is quite thorough and insightful.


Weaknesses:
* Several sections of the paper regarding the AMAS data synthesis are not explained very clearly. The entire Tool Mocker section is very confusing, and it is unclear what purpose this component has in the framework. It isn't in Algorithm 1 and it is very unclear what exactly this component does, and when is it used. Figure 1 seems to imply that it has some kind of "hybrid strategy" which is completely unexplained in the rest of the paper, and more over, what even is "Online Models" or "LLM Simulation"? The Explorer is also very confusing to read in Section 2.4. It says that it is used when "The instruction queue Q is exhausted", so when the queue is exhausted, it transitions completely to synthetic user intents? How are these intents sampled? How does it analyze the current trajectory, why does it need to do this?
* The baseline "Serial" is also not very sell explained. It just says that the "asynchronous distributed execution" is replaced with "serial receiving and dispatching process". It is unclear what exactly this means.
* There is no ablation regarding the Tool Mocker or the The Explorer. It is unclear how exactly these components affect performance in Table 1.
* In Table 1, it is also unclear how fair these comparisons are. In Section 4.1, it states that 12k samples were used to train the model, were 12k samples also used from the Origin baselines.
* The paper has this Case Study in Section 4.4 where their AMAS technique was able to respond that one of the tools can only handle 24h weather forecasts, but it is unclear why exactly AMAS does this. This capability is not brought up in the rest of the paper. In fact, in Section 4.1, it says that they filtered out trajectories with unrecoverable failures, wouldn't this be a unrecoverable failure, so then wouldn't you expect AMAS to fail at this? In order to better understand this phenomena, it would be better to construct a test set of tools and intents where the task is impossible and then report how often the model reports this impossibility, and how often it hallucinates.
* The data synthesis here initializes its dataset based on three existing datasets, ToolACE, xLAM, and TOUCAN, so does this mean that this technique relies on some existing data already existing? Given a completely new set of tools and tasks, how would this data generation handle this?
* In Section 2.1, the Overview, the authors "propose a centralized planning and distributed execution strategy" to handle multi-agent frameworks. However, this kind of strategy is fairly well known and commonplace, for example [1]. The related work and this overview section should be amended with more citations to prior work in this area.
* The overall grammar and formatting of the paper is lacking and could be improved across the board. For example, in Figure 3, the dataset TOUCAN is misspelled "TOUNCAN". There are also a lot of missing periods in the paper.
* One question I have is in Figure 3, where it seems to indicate that the Hallucination Rate of TOOLACE is around 20%, are the authors implying that the dataset from ToolAce itself has a 20% hallucination rate in the trajectories themselves?
* What is the difference between Think and Non-Think in Figure 3? It seems like this is not the same as the Think vs non-Think described in the ablation in Section 4.3.


[1] Shen, Yongliang, et al. "Hugginggpt: Solving ai tasks with chatgpt and its friends in hugging face." Advances in Neural Information Processing Systems 36 (2023): 38154-38180.

---

> ### Author Rebuttal · Authors · 2026-03-27
>
> **W1**:
>
> Please refer to the full technical description of AMAS framework in rebuttal to reviewer Gy3i before preceeding with the rest of this reply, due to rebuttal characters limits.
>
> **W2**:
>
> The Serial baseline means only changing AMAS's asynchronous execution to serial execution: Organizer distributes all subtasks at once and then blocks, waiting for all results to be returned before proceeding to the next planning. Otherwise, it is the same as the AMAS method. By contrast, Serial lacks immediate response to intermediate execution errors and is less efficient in context management.
>
> **W3**:
>
> Please see rebuttal to reviewer Gy3i.
>
> **W4**:
>
> As mentioned in Sec 4.1, queries used all come from the same items in three original datasets, which means "the exact same query IDs". All SFT experiments used dataset synthesizing on these queries, differing only in synthesis method (AMAS Async and Serial) or the original response (Origin).
>
> **W5**:
>
> 1. As shown in Figure 9, the user requested "mid-December weather" beyond the tool's capabilities. The original data ignored four tool errors and fabricated NOAA data. As for AMAS, when Executor's weather tool returned an anomaly of "only supports 24-hour forecasts" (red X in the figure), Organizer incorporated this information into global context with replanning, determined that task was uncompletable, and informed limitation with a rejection decision.
>
> 2. Regarding data filtering, this is because Organizer' s (Qwen3-Next) long context capabilities are still not perfect. We observed that Organizer sometimes allocated many failed subtasks due to retries timeout and format. We manually removed these by detecting outliers in number of tool calls. It should be noted that there is no fully automated model-based method can guarantee perfect data synthesis. We aim to demonstrate that AMAS framework has inherently high quality, efficiency and robustness compared to manually configured cleaning and filtering pipelines, rather than directly generating perfect data in one run which is unrealistic.
>
> 3. As.suggested, we constructed 100 simple impossible cases including querying future information and tasks beyond capabilities of tools, covering different complexities. Term "base" denotes a single vanilla Qwen3-4B.
>
> |&emsp;&emsp;category|&emsp;&emsp;&emsp;&emsp;setting|count|AMAS handled|base handled|
> |:-:|:-:|:-:|:-:|:-:|
> |weather/stock/news|1-5 tools,1-4 turns|60|60|28|
> |insufficient tool|1-4 tools,1-2 calls,1-5 turns|40|40|18|
>
> **W6**:
>
> Please see rebuttal to Reviewer Gy3i. Additionally, for one new sample, AMAS requires a tool definitions ($T$) in JSON format and an initial queue of user queries in one conversation ($Q$). The output is responses to the user queries forming the full conversation.
>
> **W7**:
>
> We agree with reviewer's observations that this design has indeed been extensively studied, and have discussed related scaling principles in the Introduction [1]. Furthermore, we have noticed industrial practices such as agent swarm [2]. We will include these alongside with reviewer's suggestion in relevant citations.
>
> The core contribution of AMAS lies in its data synthesis paradigm rather than a simple application of general multi-agent protocol. Traditional methods rely on manual filtering and discard erroneous but correctable data, whereas AMAS autonomously generate trajectories with success paths, self-correction, and boundary recognition. To the best of our knowledge, we are the first to demonstrate benefits of this paradigm in data synthesis, including advantages in contextual efficiency, data diversity, and robustness to models and tools itself, and have conducted thorough analysis and verifications.
>
> **W8**:
>
> We will modify these typo.
>
> **W9**:
>
> The principle for judging hallucination is: "Number of occurrences where tool names do not exist or parameters are incorrect", as also available in Appendix A.3. We calculated the percentage of hallucinations out of all conversations, it showed that the hallucination rate for ToolACE is around 20%. We recommend additional quality checks when using this dataset in the future.
>
> **W10**:
>
> In Figure 3, Think and Non-Think refer to whether thinking content is included when judging the number of hallucinations and exceptions handling. The reason for different results is that thinking content might provide explicit evidence of corrective intent. For example, if Organizer has considered that the failure is only temporary and directly performed a retry, ignoring thinking content will lead to inconsistent judgments about the nature of trajectory.
>
> The ablation in Table 2 refers to whether loss is masked for thinking content during training, considering that Qwen3-4B supports both enabling and disabling thinking.
>
> [1] Kim, Y., et al. Towards a science of scaling agent systems. arXiv preprint arXiv:2512.08296, 2025b.
> [2] Kimi Team, et al., "Kimi K2.5: Visual Agentic Intelligence," arXiv preprint arXiv:2602.02276, 2026.

---

> > ### Author Rebuttal · Reviewer_1LcP · 2026-04-03
> >
> > I thank the authors for their rebuttal.
> >
> > W1: In the Rebuttal to Gy3i, it is written:
> > ```
> > The hybrid tool mocker handles two distinct scenarios as specified in the system prompt:
> >     "Online Models": For simple, real-time queries such as time and weather, the actual API is invoked.
> >     "LLM Simulation": For complex or custom services, it acts as a sandbox simulator that returns outputs and ensures that erroneous parameter commands trigger simulated error messages.
> > ```
> > So, I am looking at the system prompt in Appendix A.4 and it seems like this is a prompt which just simulates plausible responses from the api, so is this the system prompt for the LLM Simulator, or the Hybrid Tool Mocker?
> >
> > W4: My question was mainly about the dataset size i.e. how many datapoints were generated by each method, and whether or not this was consistent across the baselines. Essentially, if you start with the same queries, do you get the same number of samples in your dataset, will the model see the same number of gradients?

---

> > > ### Author Response · Authors · 2026-04-03
> > >
> > > We are pleased to see the further progress made in reaching an agreement and appreciate your patience throughout the review process. We have provided more detailed explanations below for the remaining questions, and hopefully they might fully address your concerns.
> > >
> > > **W1**
> > >
> > > Prompt in A.4 is the system prompt for the Hybrid Tool Mocker. As stated in lines 689-692:
> > >
> > > > Ignore your actual tool availability, MCP protocol compliance, or the validity of IDs/Keys. If the user asks, you MUST hallucinate a plausible, high-fidelity JSON response populated with rich, invented data. However, when real-time information is required, utilize external tools to fetch accurate data rather than inventing it.
> > >
> > > This instructs mocker to dynamically switch strategies: it should prioritize generating high-fidelity mock returns for general scenarios, acting as an LLM simulation without relying on any actual tools. However, when a tool request involves real-time information retrieval, the mocker utilizes a conditional fallback path to execute these tools, corresponding to Online Models.
> > >
> > > We predefined the following real-time tool functions as below. Please note that these are simpler examples and are not identical to what we used in practice:
> > >
> > > **server side**
> > >
> > > ```
> > > openapi: 3.1.0
> > > info:
> > >   title: Mini Tool Server
> > >   version: 1.0.0
> > >
> > > paths:
> > >   /time:
> > >     get:
> > >       summary: Get current time
> > >       parameters:
> > >         - in: query
> > >           name: timezone
> > >           required: true
> > >           schema: { type: string }
> > >       responses:
> > >         "200":
> > >           description: OK
> > >
> > >   /news:
> > >     get:
> > >       summary: Search latest news
> > >
> > >   /news:
> > >     get:
> > >       summary: Search latest news
> > > ```
> > >
> > > **mocker side**
> > >
> > > ```
> > > async def get_time(base_url: str, timezone: str) -> dict:
> > >     async with httpx.AsyncClient() as client:
> > >         r = await client.get(
> > >             f"{base_url}/time",
> > >             params={"timezone": timezone},
> > >             timeout=10.0,
> > >         )
> > >         r.raise_for_status()
> > >         return r.json()
> > >
> > > async def get_weather(base_url: str, lat: float, lon: float) -> dict:
> > >
> > > async def get_news(base_url: str, q: str) -> dict:
> > > ```
> > >
> > > Furthermore, these tools are defined in OpenAI tool-calling format and added to the mocker's input. Considering that real tools contain sensitive information like actual server used, accounts and URLs, we have not listed specific function declarations in the paper. We will include a full de-identified example in Appendix.
> > >
> > >
> > > **W4**
> > >
> > > Yes, exactly. Each method includes a consistent 10,000 generated datapoints for fine-tuning. We apologize for having to limit the length of explanation in the previous rebuttal due to the character limit. We hereby detail the full life cycle of the data synthesis:
> > >
> > > 1. We collected three existing comprehensive tool-calling datasets. Each data sample is a single-turn or (in fact mostly) multi-turn conversation with its own tool list.
> > > 2. For each sample, we removed responses for every turn and kept the input of that turn as the instruction. So we obtained an instruction queue, corresponding to $\mathcal{Q}$ in Algorithm. The following descriptions all pertain to a single $\mathcal{Q}$.
> > > 3. We employed different methods, including AMAS (Async) and AMAS (Serial). For each instruction in $\mathcal{Q}$, denoted as $q$, the Organizer starts planning $q$ and splits it into multiple subtasks. Executors asynchronously perform them one by one and return, then Organizer re-plans. This process iterates until Organizer determines an answer to $q$, denoted as $m_{finished}$. Now we complete the current turn as also showned in Algorithm.
> > > 4. We then proceed to solve the next $q$ in $\mathcal{Q}$, at which point all components are reset. This continues until $\mathcal{Q}$ is exhausted. The optional Explorer will generate a new $q$ based on the previous history and then activate the system to solve it once again. Finally, we obtain a fully synthesized conversation data sample. Noted that Origin simply means keeping the initial responses. Given that Origin datasets themselves have already undergone multiple stages of synthetic processing, so the setup effectively serves as a standard and valid baseline for comparison.
> > >
> > > As we mentioned "queries used all come from the same items in three original datasets, which means 'the exact same query IDs'" in previous rebuttal, it means we utilized the exact same set of datasource from Origin, removed responses in each turn for every sample and kept instructions (queries in another words), and then synthesized responses to form full conversations using different methods. Therefore, all experiments have the exact same volume of data and identical queries each (gradients in your term), ensuring that the comparison is completely fair.
> > >
> > > We look forward to your feedback and hope that our responses help you better understanding of the paper and do not bring additional burden.

---

### Official Review · Reviewer_Gy3i · 2026-03-13

**Soundness:** 2
**Presentation:** 2
**Significance:** 2
**Originality:** 2
**Overall Recommendation:** 4
**Confidence:** 3

**Summary:**

This paper proposes AMAS (Asynchronous Multi-Agent Synthesis), which is a hierarchical framework that uses asynchronous task scheduling to simulate realistic tool interactions. Furthermore, AMAS introduces an Explorer to generate diverse queries and a Tool Mocker to simulate tool responses and failures. Experiments show that data generated with AMAS improves robustness, diversity, and context efficiency, and that fine-tuning Qwen3-4B on the synthesized trajectories yields better performance on benchmarks such as BFCL, demonstrating improved tool-use reasoning and error-handling capabilities.

**Compliance With Llm Reviewing Policy:**

Affirmed.

**Final Justification:**

The author's response has resolved my questions.

**Key Questions For Authors:**

- What’s the accuracy of the LLM-based mock server?

**Limitations:**

yes

**Strengths And Weaknesses:**

Strength:
- This paper explores an important research direction of synthesizing high-quality multi-turn trajectory training datasets.
- The evaluation contains a detailed, in-depth analysis of the quality of the synthesized dataset.

Weakness:
- The effectiveness of the Explorer is not evaluated in detail. It is important to conduct more in-depth analysis and case studies to justify this design.
- The training experiment is not comprehensive enough. Currently, it only contains one model (i.e.,  Qwen3-4B), making the results less convincing and generalizable to more general settings. More diverse LLMs from different model families with larger sizes should be included in the training experiment.
- More in-depth analysis is required for the training experiment. Since one of the major advantages of using AMAS to synthesize trajectory data is reducing the Hallucination/Hallucination Handled/Exception Handled Rate inside the training set, it’s important to show that the model trained on this kind of dataset can also learn such behavior efficiently.

---

> ### Author Rebuttal · Authors · 2026-03-27
>
> **W1** & **W2**:
>
> 1. We provide ablations across model families, sizes and the Explorer. Terms 'w/ E' and 'w/o E' denote data with and without Explorer, respectively.
>
> |**Benchmark**|&emsp;**Setting**|**Qwen3-4B**|**Qwen3-30B-A3B-Instruct**|**Llama-3.1-8B-Instruct**|**OLMo-3-7B-Instruct-SFT**|**OLMo-3-1125-32B-Think-SFT**|
> |:-:|:-:|:-:|:-:|:-:|:-:|:-:|
> |BFCL\_v3|Base|57.6|58.6|49.6|48.9|57.0|
> ||w/ E|**59.6**|**61.2**|**55.8**|**51.5**|**59.3**|
> ||w/o E|59.5|61.0|53.2|51.5|58.0|
> |$\tau^2$-Retail|Base|31.1|57.0|5.3|33.6|60.2|
> ||w/ E|**36.9**|58.2|**6.6**|**39.2**|63.7|
> ||w/o E|36.8|**60.1**|6.3|38.6|**65.4**|
> |$\tau^2$-Airline|Base|17.2|38.7|38.6|29.3|40.8|
> ||w/ E|**29.8**|**42.6**|**39.7**|**29.3**|**41.9**|
> ||w/o E|26.8|40.2|39.6|28.1|41.3|
> |$\tau^2$-Telecom|Base|13.4|12.3|2.6|13.2|21.2|
> ||w/ E|**18.3**|**25.4**|**7.3**|13.0|21.9|
> ||w/o E|17.2|25.4|6.8|**13.2**|**22.6**|
>
> The result consistently show that training on AMAS data with Explorer improved overall performance by 1% to 3% compared to w/o Explorer, surpassing base models by a margin. This demonstrates that the diversity brought by Explorer can indeed improve data quality.
>
> 2. We evaluate diversity score with G-Vendi score used in Sec 3.4.
>
> |G-Vendi|ToolACE|xLAM|TOUCAN|
> |:-:|:-:|:-:|:-:|
> |w Explorer|68.7127|65.3558|118.7195|
> |w/o Explorer|68.0881|46.9200|118.2643|
>
> It shows that data with Explorer scores higher on the G-Vendi Score, thus giving wider coverage of training samples in the gradient space.
>
>
> **W3**:
>
> 1. As in Table 1 in Sec 4.2 , it can be seen that model trained on AMAS data shows the most significant improvement in BFCL subitems like Irrelevance and Agentic Memory by 4% to 7%, which require explicit rejection or state recognition capabilities.
> 2. We additionally calculated the percentage of trajectories which performed self-correction when evaluating base and sft Qwen3-4B on BFCL v3.
>
> ||samples|calls|error|correction|error ratio(%)|correction ratio(%)|
> |:-:|:-:|:-:|:-:|:-:|:-:|:-:|
> |base|4441|9694|2924|41|30.16|1.4|
> |sft|4441|8031|1896|176|23.61|9.3|
>
> The result also confirmed that trained model exhibited better error correction and hallucinations handling patterns.
>
> **W4**:
>
> We evaluated 1000 samples for the success rate of tool mocker. We used k2.5 for labeling exceptions, handled cases and average exception ratio per conversation.
>
> ||calls|exceptions|handled|avg exception ratio(%)|
> |:-:|:-:|:-:|:-:|:-:|
> |ToolACE|5188|223|223|1.8|
> |xLAM|2171|34|34|1.64|
> |TOUCAN|9009|1674|1531|13.12|
>
> This shows that the mocker exhibits a certain natural failure rate, which depends on instructions in the dataset. AMAS recognizes this very well, and these failures trigger the Organizer to asynchronously replan and achieve a high handled rate.
> Specifically, tool mocker generated more exceptions on TOUCAN dataset, as it involves many MCP tasks. Despite the role specification in system prompt, the LLM-based mocker sometimes explicitly rejects requests by saying it doesn't possess the actual MCP tools. However, the sufficiently high handle rate further demonstrates the error correction capabilities provided by the framework.
>
> **Additional parts**
>
> We clarify the AMAS component design here, as reference for all the review process.
>
> 1. The system consists of an Organizer for decentralized planning and Executors for subtask execution based on atomic tool-calling. At each round of conversation begins, Organizer maintains a context summary and distributes subtasks within this turn, replanning instantly based on asynchronous feedback from Executors. This design achieves native resilient error correction that when a tool call fails, Organizer can directly intervene and correct the current trajectory without hard-coding. They will be both reset when entering the next round of conversation.
>
> 2. The hybrid tool mocker handles two distinct scenarios as specified in the system prompt:
>     *   "Online Models": For simple, real-time queries such as time and weather, the actual API is invoked.
>     *   "LLM Simulation": For complex or custom services, it acts as a sandbox simulator that returns outputs and ensures that erroneous parameter commands trigger simulated error messages.
>
>     Regarding line 13 of Algorithm 1 (`Trigger e(t)`), the executor $e$ sends the call to the tool mocker. We will clarify this interaction in the updated version of the algorithm.
>
> 3. We collected three existing datasets and kept only queries as data seeds, and used AMAS framework to synthesize responses. We employ an optional and independent Explorer that uses previous context summaries and user input from the last round to simulate a real user asking further questions at the end of conversation, see the prompt in the appendix. By doing so, Explorer can avoid generating overly simple queries or queries that have already been executed previously, ensuring both increased dialogue diversity and quality.

---

> > ### Author Rebuttal · Reviewer_Gy3i · 2026-04-04
> >
> > Thank you for the response. It has resolved my questions.

---

> > > ### Author Response · Authors · 2026-04-04
> > >
> > > Thanks for your positive feedback and we are glad that our responses have addressed your concerns. We will incorporate all above results into the paper.

---

### Official Review · Reviewer_Tbp2 · 2026-03-30

**Soundness:** 3
**Presentation:** 3
**Significance:** 3
**Originality:** 3
**Overall Recommendation:** 4
**Confidence:** 4

**Summary:**

This paper presents AMAS, an asynchronous multi-agent data synthesis framework based on a two-layer architecture of organizer and executor, for generating high-quality training data for LLM tool calls. The core motivation of this framework is that existing data synthesis methods have two fundamental flaws: the static pipeline discards failed trajectories, resulting in information loss, and serial execution is inefficient in complex scenarios. AMAS achieves the decoupling of centralized planning and distributed execution through asynchronous event loops and the wait-for-any mechanism, enabling the system to dynamically re-plan when the tool returns an exception, converting the discarded erroneous trajectories into self-correcting training signals. The framework also introduces the Explorer module to alleviate seed bias and the Tool Mocker to simulate the generation of natural errors by the tool server. The authors conducted data quality analysis on three datasets, ToolACE, xLAM, and TOUCAN, and used synthetic data to fine-tune Qwen3-4B for SFT. They verified the improvement in downstream task performance on BFCL and τ²-bench.

**Compliance With Llm Reviewing Policy:**

Affirmed.

**Final Justification:**

The rebuttals have resolved my questions.

**Key Questions For Authors:**

See Weekness.

**Limitations:**

yes

**Strengths And Weaknesses:**

The paper identified a practical and significant issue - the "success bias" of the current tool invocation dataset is indeed a key bottleneck that restricts the robustness of the model. The idea of regarding error trajectories as training resources rather than waste is enlightening. The framework design is logically clear. The double-layer architecture of Organizer-Executor and the asynchronous event loop mechanism have reasonable engineering motivations. The formal description of Algorithm 1 is also convenient for replication. The data analysis dimension is relatively comprehensive, covering the distribution of tool invocations, hallucination rate, exception handling rate, G-Vendi diversity, and context efficiency of the Pareto frontier, from multiple angles to characterize the characteristics of synthetic data. The consistency verification of manual annotation and automatic evaluation (r = 0.896) enhances the credibility of the evaluation protocol. The case analysis (Figure 9 and Appendix F) intuitively shows the behavioral differences between Origin and AMAS when facing tool failures, and is quite persuasive.


**Question 1**: The paper regards "transforming error trajectories into self-correction training signals" as one of its core contributions. It is mentioned in Section 4.1 that rejection sampling was used to filter out irrecoverable failures in the synthetic data, and up-sampling was applied to the trajectories that were successfully corrected. However, an ablation experiment to directly verify this contribution was lacking in the experiment. Specifically, if a training set containing only Error-Free trajectories (excluding all Error-Recovered samples) was compared with the complete training set, the marginal contribution of self-correction trajectories could be directly quantified. The current results in Table 1 cannot distinguish whether the performance improvement is due to the asynchronous mechanism improving the data distribution itself or the additional training signals provided by the correction trajectories. This makes the paper's most core assumption lack direct experimental support.

**Question 2**: The downstream SFT experiment only uses the Qwen3-4B single model and 12,000 training samples. This experimental scale is insufficient to support the paper's claim of the framework's generality. Considering that Organizer used Qwen3-Next-80B-A3B-Thinking during the synthesis process, Executor and Tool Mocker used Qwen3-4B-Instruct, and the fine-tuning target is also Qwen3-4B, the entire experiment is always running in the Qwen series model loop. If the conclusion is verified on different model families (such as LLaMA) or with different parameter scales (such as 7B, 14B), will the conclusion still hold? Moreover, of the 12,000 samples, only 10,000 were from AMAS synthesis. How were these samples distributed across the three source datasets and what proportion did each account for? The paper does not clarify this.

**Question 3**: The baseline comparison in the paper is limited to Origin (original data) and Serial (serial variant), which is essentially an internal comparison of the framework's variants, lacking a horizontal comparison with existing similar data synthesis methods (such as AGIgen-MT, ToolACE-MT, EigenData). In particular, EigenData also adopted a hierarchical multi-agent architecture for tool invocation data synthesis and achieved significantly higher results on τ²-bench, but the paper neither cited nor discussed this. Comparing only with its own variants is prone to selection bias and cannot demonstrate the true advantage of AMAS over existing technical solutions.

**Question 4**: The paper proposed two auxiliary modules, Explorer and Tool Mocker, but the ablation experiment (Table 2) only covered the dimension of Thinking vs No Thinking. How significant is the specific contribution of the Explorer module to data diversity (enabled vs disabled)? How does Tool Mocker generate natural errors compared to manually injected errors or using real APIs? What is the impact of the number of asynchronous executors on the synthesis quality and efficiency? These are key experiments to understand the marginal contribution of each component of the framework, but they are not covered. The exact source of the G-Vendi score improvement in Figure 7, whether it comes from the Explorer's introduction of new queries or the path diversity brought by the asynchronous mechanism itself, cannot be determined at present.

**Question 5**: On the Telecom subset of τ²-bench, Async (17.62) only improved by 0.12 percentage points compared to Base (17.5), while Serial (18.20) was higher than Async. This contradicts the paper's claim that the asynchronous framework is overall superior to the serial method. However, no discussion was made in the main text regarding this. Also on the BFCL-v3 Live subset, Origin (75.19) was higher than Async (74.78), but this was not mentioned either. Additionally, all experimental results were point estimates from a single run and no standard deviations or confidence intervals were reported. Considering the sensitivity of SFT training to random seeds, it is impossible to confirm whether the 1-2 percentage point difference between Async and Serial in most cases is statistically significant.

**Question 6**: The paper repeatedly emphasizes the efficiency advantages of the asynchronous mechanism in Section 1 and Section 3.3, but the actual efficiency evidence only comes from the Pareto front graph in Figure 6, which uses the scratchpad length as an indicator, rather than the actual computational costs. The paper did not report the actual running time, total token consumption, or API call count of Async vs Serial vs Origin when synthesizing the same number of samples. Considering that the asynchronous framework requires additional maintenance of state management, event loops, and dynamic scheduling logic, can the engineering overhead of this be masked by the scratchpad length indicator? If each scratchpad step of Async involves more parallel token generation, then a shorter scratchpad length does not necessarily mean a lower total computational cost.

---

> ### Author Rebuttal · Authors · 2026-03-31
>
> We sincerely thank the reviewer for providing all above under a tight schedule. We have carefully addressed each raised below.
>
> **Q1**:
>
> We resampled 10000 error-free trajectories and randomly sampled the same amount from full synthetic datasets. Statistical metrics across three evaluation runs are reported.
>
> |benchmark|category|error-free|full sampling|Async|
> |-|-|-|-|-|
> |BFCL v3|live|73.29±0.052|74.60±0.045|74.78|
> ||non-live|61.59±0.119|63.11±0.141|63.24|
> ||multi-turn|24.95±0.184|25.54±0.087|25.46|
> |$\tau^2$-bench|Retail|32.26±0.564|33.39±0.406|35.76|
> ||Airline|23.84±2.348|25.08±2.362|27.81|
> ||Telecom|17.53±1.009|17.73±0.303|17.62|
>
> Results on Qwen3-4B show comparable performance between full and error-free training while the latter shows slightly lower scores, demonstrating the necessity of correction trajectories.
>
> **Q2**:
>
> Please refer to rebuttal to reviewer Gy3i regarding experiments on model families and sizes. Sampling strategy was motivated by balancing tool calls frequency and conversation turns. The statistics is as followed:
>
> |source|ToolACE|xLAM|TOUCAN|
> |-|-|-|-|
> |count|5000|2000|3000|
>
> **Q3**:
>
> "Origin" itself is the result of a systematic data synthesis method, thus it is fair to compare it with AMAS on framework level. Regarding new benchmarks, we do not find AGIgen-MT via online searching, but there is actually one similar paper named APIGen-MT. Its predecessor, APIGen, is exactly the xLAM dataset that we used and compared in Origin. For others, ToolACE-MT introduces a non-autoregressive three-stage tool synthesis framework. On the one hand, to our knowledge, it is not open source; on the other hand, we also compared AMAS with TOUCAN, which is an industrial-level example of this type of pipeline for tool calling synthetic datasets with larger data volume and richer sources. EigenData was published on Arxiv on March 5, 2026, which is later than submission deadline, so it was feasible that we could not discuss or cite it. We are welcome to add the comparison of the above three with AMAS in the next round if necessary indeed. We are also happy to further discuss works from the same period this year.
>
> **Q4**:
>
> Please refer to rebuttal to reviewer Gy3i. To clarify, as also explained in sec 2.6, tool mocker's behavior stem from its LLM role-playing capability, functioning as generating standard error outputs when receiving clearly wrong parameters, enhancing diversity and time efficiency compared to real APIs. Executors count in AMAS is dynamically determined by Organizer to minimize atomic task execution. We provide the average executors count statistics here:
>
> |source|ToolACE|xLAM|TOUCAN|
> |-|-|-|-|
> |count|2.95|2.05|7.07|
>
> **Q5**:
>
> We thank the reviewer for pointing out omissions in our initial draft and have incorporated these valuable suggestions. $\tau^2$-bench is challenging for Qwen3-4B in multi-turn interactions, modest gains are reasonable without aggressive optimization for instruction following. As for Serial sometimes outperforming Async, it suggests model benefits more from correct tool selection than efficient context history management. Though Origin trained model performs better on live, Async achieves superior results on v3 overall, especially in multi-turn. To address the statistical point, mean±std over three runs is reported:
>
> |benchmark|category|avg|base|
> |-|-|-|-|
> |BFCL v3|live|74.68±0.070|72.19|
> ||non-live|63.53±0.233|61.32|
> ||multi-turn|25.07±0.341|11.75|
> |$\tau^2$-bench|Retail|35.32±0.145|28.1|
> ||Airline|27.27±0.901|12.0|
> ||Telecom|18.69±0.759|17.5|
>
> **Q6**:
>
> We appreciate this important suggestion. Scratchpad refers to context summary for efficient token management, see Algorithm and Appendix please. Using scratchpad as a proxy for context processing efficiency is reasonable, as token length alone relie on model type, and time costs vary by deployment hardware (e.g., models with more concise reasoning could significantly reduce token overhead). It is indeed meaningful to compare baselines under the same framework and model.
>
> Regarding baselines, we cannot report Origin's cost, as they completed data synthesis without reporting costs or reproducible metrics with especially ToolACE omits model types and costs. Some data underwent manual inspection, which is beyond our scope to measure. In contrast, our fully automated approach brings natively data of high quality without requiring human intervention. Nevertheless, we do believe reporting cost is essential. We resynthesized 1000 AMAS samples and report average per sample runtime and token usage. Noted that including thinking significantly has much more tokens.
>
> |source|method|time(s)|tokens|
> |-|-|-|-|
> |ToolACE|Async|605.1|4876.4|
> |xLAM||312.5|2629.0|
> |TOUCAN||1172.8|7336.5|
> |ToolACE|Serial|634.7|6113.4|
> |xLAM||301.0|2815.5|
> |TOUCAN||4784.9|7912.6|
> |ToolACE|Origin|/|360.6|
> |xLAM||/|80.6|
> |TOUCAN||/|4718.4|
>
> It shows Async achieves lower cost than Serial, aligning with higher efficiency as discussed in the paper.

---

> > ### Author Rebuttal · Reviewer_Tbp2 · 2026-04-02
> >
> > The rebuttals have resolved my questions and I have raised the score.

---

> > > ### Author Response · Authors · 2026-04-03
> > >
> > > Thank you for your positive feedback, and we appreciate the opportunity to clarify your concerns. We will add the new content to the manuscript.

---

### Decision · Program_Chairs · 2026-04-30

**Decision:**

Reject

**Comment:**

AMAS proposes a hierarchical asynchronous multi-agent framework for tool-calling data synthesis that converts error trajectories into self-correction training signals, with solid data quality analysis and consistent downstream improvements. Scores are 4/4/3; two reviewers fully resolved after rebuttal added cross-model ablations (LLaMA, OLMo, Qwen at multiple scales), Explorer ablations, and cost/latency breakdowns. Reviewer 1LcP remains at weak reject with lingering concerns about method clarity and reproducibility that are legitimate — the Tool Mocker and Explorer descriptions required rebuttal clarification that should have been in the paper.